# Simulation of pH-Dependent Conformational Transitions in Membrane Proteins: The CLC-ec1 Cl^−^/H^+^ Antiporter

**DOI:** 10.3390/molecules26226956

**Published:** 2021-11-18

**Authors:** Ekaterina Kots, Derek M. Shore, Harel Weinstein

**Affiliations:** Department of Physiology and Biophysics, Weill Cornell Medical School, New York, NY 10065, USA; edk4002@med.cornell.edu (E.K.); des2037@med.cornell.edu (D.M.S.)

**Keywords:** CLC transporters, pH-dependent activity, molecular dynamics (MD) simulations, high-resolution atomic force microscopy (HR-AFM), protonation state representations, net charge conservation protocol

## Abstract

Intracellular transport of chloride by members of the CLC transporter family involves a coupled exchange between a Cl^−^ anion and a proton (H^+^), which makes the transport function dependent on ambient pH. Transport activity peaks at pH 4.5 and stalls at neutral pH. However, a structure of the WT protein at acidic pH is not available, making it difficult to assess the global conformational rearrangements that support a pH-dependent gating mechanism. To enable modeling of the CLC-ec1 dimer at acidic pH, we have applied molecular dynamics simulations (MD) featuring a new force field modification scheme—termed an Equilibrium constant pH approach (ECpH). The ECpH method utilizes linear interpolation between the force field parameters of protonated and deprotonated states of titratable residues to achieve a representation of pH-dependence in a narrow range of physiological pH values. Simulations of the CLC-ec1 dimer at neutral and acidic pH comparing ECpH-MD to canonical MD, in which the pH-dependent protonation is represented by a binary scheme, substantiates the better agreement of the conformational changes and the final model with experimental data from NMR, cross-link and AFM studies, and reveals structural elements that support the gate-opening at pH 4.5, including the key glutamates Glu_in_ and Glu_ex_.

## 1. Introduction

Members of the CLC family of transmembrane proteins that mediate voltage-dependent transport of chloride across the membrane cell offer powerful examples of the key role that the ambient pH can play in the mechanisms of function of biomolecular systems in the cell membranes. The CLCs family comprises both ion channels and transporters that share major structural motifs [1,2]. Nine different human CLCs were identified, expressed in brain, heart, liver, gut and muscle tissues [3,4]. They have major physiological roles in maintaining homeostasis (e.g., in skeleton muscle tissues, the CLC-1 channel supports action potential generation by sodium and potassium re-polarizing the muscle fiber) [5,6]. The CLC transporter members of the family operate as Cl^−^/H^+^ antiporters where the intra- and extracellular concentrations of permeant Cl^−^ and pH levels regulate conductivity. The prokaryotic analog of human CLC transporters, CLC-ec1, has been studied extensively as a mechanistic and structural prototype for the human transporters in this class [7,8,9,10].

CLC-ec1 shares the common structural architecture of a homodimer, in which each subunit has an independent anion-permeation pathway [11,12]. Structure–function studies have identified key residues involved in the transport mechanisms, such as the two glutamate residues, Glu_ex_ and Glu_in_, located, respectively, in the extracellular and intracellular sides of each protomer [13,14,15]. Thus, Glu_ex_ was proposed to act as both a “gate” for chloride transport and a mediator for proton binding [9,16], whereas Glu_in_ is considered to be a proton transfer site [10,17].

Remarkably, no conformational rearrangements other than reorientation of the Glu_ex_ sidechain had been observed in crystallographic structure determinations of CLC-ec1, even in a wide range of pH < 9 to as low as pH 5.5 [18] (see also Appendix A). This changed with the determination of an X-ray structure of the triple CLC-ec1 mutant E148Q/E203Q/E113Q–PDB ID 6V2J [18]. This mutant, referred to as QQQ, was shown to share the main functional and structural properties of the wild-type CLC-ec1 at acidic pH [18] and exhibits ~2 Å conformational changes in the positions of the N, O, P and G helices. These findings are supported by results from DEER experiments, fluorescence, NMR, and cross-linking studies [18]. 

Most recently, investigations with a novel High-Resolution AFM (HR-AFM) method [19] captured conformational changes occurring at pH 4.5 in the extracellular regions of CLC-ec1 that protrude above the membrane. The changes were shown to relate to the proposed structural changes associated with gate opening [19]. The identification of the protruding residues of CLC-ec1 that were sensed by the high-resolution AFM method, and the interpretation of the underlying structural changes translated into the peaks observed in HR-AFM maps were aided by results from conventional (canonical) molecular dynamics (cMD) simulations of CLC-ec1 that we carried out at pH 4.5 and 7.6. To this end, we had created separate models of the dimeric CLC-ec1 system for each pH value, which differed in their assignment of fixed protonation states of titratable residues based on individual pKa values relevant to the range between pH 4.5 and pH 7.6. These canonical MD simulations were unable to produce different ensembles at the two pH values in ~10 µs of MD simulations [19].

The main reasons for this failure of cMD to produce the information about the different conformational ensembles of the CLC-ec1 at the two relevant pHs are (i) the use of fixed protonation states at each pH in the force-field formalism of cMD, and (ii) the binary nature of the proton occupancy representation in the force field, which then provides an appropriate model of the protonation state only at the two extremes of the pH range. To remedy these shortcomings and model pH-dependent conformational transitions in the CLC-ec1 protein at pH 4.5 where the structure has not been determined, we have developed a new force-field scaling approach that we termed Equilibrium Constant pH (ECpH). The formal theoretical framework of ECpH is described in detail [20] and is briefly outlined in Methods. The ECpH protocol enables MD simulations within a canonical (or N, P, T) ensemble starting with a protein construct in which all H^+^ titratable sites are fully protonated, and the electrostatic and Van der Waals interactions for each protonatable site are scaled according to the ambient pH and its user-defined pKa value. We show here that application of ECpH simulations of the dynamics of the CLC-ec1 transporter at various pH values do indeed reveal the conformational changes underlying the observations from the HR-AFM measurements [19] and agree with observations from a variety of methods including NMR and cross-linking experiments. This progress is due to the ability of the method to describe changes in the protonation probability of the titratable residues from an ensemble definition of protonation probability. The specific changes in protonation probability are produced by the ensemble of local and global conformational changes of the protein at different pH values, thus enabling ECpH to provide a useful construct of the system at a certain pH.

## 2. Results

The structural rearrangements underlying the pH-dependent transition of the CLC-ec1 dimer between opened and closed gate states observed with HR-AFM [19] and a variety of other experimental approaches [18,21,22,23,24] were investigated with MD simulations employing the ECpH force field modification scheme. This approach enables ECpH to overcome the binary representation of the protonation states in the force field of cMD, thereby introducing a crucial improvement in the ability of MD simulations to represent protein molecules in a range of physiological pH (Appendix B). Thus, the core of the ECpH method is a linear interpolation between the protonated and non-protonated force-field parameters of each titratable residue in the system that scales the current protonation state according to its individual pKa value and environmental pH (see Section 4.1). We estimated the individual pKa values for all the titratable residues with the recently introduced PROPKAtraj [25,26] approach that revises the past core flaw of PROPKA [26] and takes advantage of the protein dynamics in its evaluation of individual pKas (see Methods). We investigated the pH-dependent structural dynamics of the CLC-ec1 macromolecular system at the physiologically relevant pH values with the equilibrium constant pH (ECpH) MD approach described in [20] and briefly reviewed in the Methods (Section 4). Additional information about the formal framework of the ECpH methodology is provided in Appendix B and Appendix C).

The simulations focused on the path of the pH-dependent CLC-ec1 conformational rearrangement that takes the molecule to the state in which the transport efficiency is largest, at ~pH 4.5. Interestingly, the comparison of crystallography data for the WT CLC-ec1 at pH 4.6 (PDB ID 1KPL) and pH 8.5 (PDB ID 1KPK) does not show major differences in the conformational state of the protein [27], but evidence from other experimental approaches has indicated conformational changes interpreted to represent the interconversion between functional states of the CLC protein in response to pH changes [18,21,22,23,24]. For example, the introduction of a disulfide crosslink by the mutations A399C/A432C was shown to inhibit Cl^−^ transport, leading to the suggestion of a connection between the resulting displacement of Helix O opening of the gate in the outward-facing state in WT protein [22]. Major structural rearrangements were also suggested from DEER experiments, cross-linking, and NMR experiments, which concluded that motions of helices N, O and P act to widen the extracellular bottleneck [18,23]. Additionally, the NMR data also suggested a pH dependence of the position of the P–Q linker that is associated with sidechain orientation of Y419 [10,24,28]. The recent high-resolution X-ray structure of a mutant construct of the CLC-ec1 protein (E148Q/E203Q/E113Q, PDB ID 6V2J), termed QQQ CLC [18], does show structural rearrangements to form an outward-facing open conformational state at pH 7.5 that mimics the WT-like conformation at low pH. The next sections describe the results from the detailed analysis of both local and global pH-dependent conformational changes in CLC-ec1, revealed by the analysis of the ECpH MD simulation trajectories.

### 2.1. Analysis of ECpH Trajectories Brings to Light the Mechanisms of Structural Changes Detected Experimentally 

The initial structure used in all the ECpH and cMD simulations was the X-ray structure of CLC-ec1 at basic pH (PDB ID 1OTS). Initial equilibration was carried out with an established protocol described in previous publications (see [18] and Methods). Starting from the equilibrated structure, parallel ECpH simulations at pH 4 and 8 were run in equivalent swarms of six replicas for ~1 µs each. The trajectories from the production runs of the simulated systems were used in the various comparative analyses described in Methods.

#### 2.1.1. Spatial Reorganization of Helices B and C

Analysis of the conformational ensembles in the ECpH trajectories at pH 4 and pH 8 revealed structural changes in the CLC-ec1 protein in response to the decrease in pH. As shown in Figure 1A, the extracellular part of helix B moved closer to the interdomain surface as indicated by the decrease in intra-subunit distance between the D73 residues from ~98 Å at pH 8 to 95 Å at pH 4. The tighter conformation shown in Figure 1A is the result of a repositioning of helices B and C. The repositioning of the R147 sidechain in helix B (Figure 1A), which follows the protonation of D54 and E148 at lower pH values, enables the rearrangement of the two helices. Another local reorientation that likely supports the helix B shift occurs near the extracellular part of the helix. Thus, at pH 4, H70 and D73 form hydrogen bonds with the backbone atoms of G66 and T71, respectively, whereas in their deprotonated form at pH 8, both H70 and D73 adopt different conformations that destabilize the alpha-helical structure at the tip of helix B (Figure 1B).

#### 2.1.2. pH-Dependent Conformational Rearrangement of Helices N, P and O

Multiple experimental studies [18,21,22,23,24] have concluded that helices N and O exhibit pH-dependent conformational rearrangements. From the ECpH trajectories, we find that the distribution of inter-subunit distances between Cα atoms of M373 (Helix N) and E385 (helix O) increases by ~4 Å at pH 4 compared to the pH 8 conformation (Figure 2), and a subtle shift of ~0.5 Å in the RMSD of their backbone atoms (Figure 2). The pH dependence of the shift in position of the N helix is likely related to the reorientation of the Glu_ex_ sidechain (E148) shown in Figure 2, accompanied by the reorganization of the hydrophobic environment located at the bottom of helix N (F190, F199, F357 and I186), as shown in Appendix A. Helix P also undergoes a significant displacement in the ECpH simulations at pH 4, as seen from the >1 Å decrease in ensemble average of the inter-subunit distance between the Y419 residues located on the P–Q loop (Figure 2). This is associated with the emergence at pH 4 of a new orientation of the Y419 sidechain which, at pH 8, is buried in the hydrophobic core of the protein (Appendix A). Notably, the position of this sidechain is similar to that observed in the crystal structure of the E113Q/E148Q/E203Q mutant CLC-ec1 (PDB ID 6V2J). Indeed, this mutant is considered to mimic the opened conformation of the WT under acidic conditions.

The extracellular parts of helices N, O and P are located in close proximity to extracellular I–J loop (H234-L249), which also changes its conformation under acidic pH conditions. This loop carries a number of titratable residues (including H234, E235 and K243) that can form pH-dependent interactions. As shown in Figure 3, at pH 4, the H234-D240 region of this loop visits a conformation that buries hydrophobic residues V236, A237, L238 and I239 in the hydrophobic core of the subunit interface, following occasional formation of hydrogen bonds between R230 and E235. At pH 8, however, a stable salt-bridge is created between R230 and E414 (P), and R230 is no longer available to orient E235 sidechain (Figure 3). Thus, the bending of the I–J loop is likely associated with the displacement of helix P and serves as an indicator of the transition.

## 3. Discussion

Application of ECpH to model pH-dependent conformational changes of the CLC-ec1 transporter produced a detailed representation of the complex structural response to the change from neutral to acidic pH. The ability of this new approach to bring to light the pH-driven conformational changes of specific residues that lead to major repositioning of transmembrane helices inferred from experimental measurements represents significant progress in the computational representation of the functional mechanism of a transporter that is activated at the acidic pH. Unlike the trajectories from canonical MD simulation with binary protonation states, the adaptation of the molecular structure to the gradual change in protonation probability modeled in the ECpH simulation framework enables the attribution of functionally important conformational rearrangements to specific local conformational rearrangements. As detailed in the results of the ECpH simulations, above, local pH-dependent rearrangements of residue interactions lead to observed rearrangements of secondary structure elements and changes in the tertiary structure of the entire protein that agree with direct experimental measurements and inferences from a variety of structure–function investigations. This holds as well for global function-related changes in the structure, as shown by comparison of our results to recently published experimental data, which identified the same changes in the switch from the inward facing conformational state at pH 7.5 to an outward facing opened state at pH 4 [23].

The ability of ECpH simulations but not the parallel cMD runs to successfully identify the local conformational changes relates to the improved model of pH-dependent conformational transitions by the force-field scaling approach. This approach enables ECpH to perform MD simulations within a canonical (or *n*, *P*, *T*) ensemble, starting with a protein construct in which the representation of electrostatic and Van der Waals interactions for each titratable site is scaled (see Section 4.2.3) according to the desired pH level and defined pKa value. In cMD, the pH-dependence is modeled by a fixed protonation configuration determined by an assumed protonation status based on the pH and pKa of a particular residue, in which the occupancy of the proton is either 1 or 0. This corresponds only to the situation at the starting and final pH. In contrast, the scaling procedure is a linear interpolation between force-field representations of protonated and non-protonated states (see Section 4.2.3) that takes advantage of an ensemble definition of protonation probability for each titratable site. This yields a construct of the system that corresponds to the statistical probability in an ensemble of the molecules at a certain pH. Here, we showed that the ECpH results agree with experimental findings about the pH-dependent rearrangement of the CLC-ec1 transporter at both the local and the global scale. Thus, the ECpH simulations revealed the details of local conformational changes produced by the evolution of protonation probabilities in the systems at one pH or another, as well as their consequence in the structural rearrangement of larger motifs and elements of secondary structures such as the transmembrane helical segments. As demonstrated by the specific comparisons of ECpH results with the corresponding experimental findings (in Section 2), the agreement obtained in the transition from basic pH to acidic pH values is remarkable at both the local and the global scale. Examples of agreement with detailed results in a recent study offer a direct comparison to parallel computations with cMD. 

The increase observed with ECpH in the inter-subunit distance measured from HR-AFM maps [19] between the highest points of helices B in the two protomers reproduces the HR-AFM maps of CLC-ec1 structure at pH 4.5 that show the displacement of the peaks attributed to the B–C linker. These changes are seen from the ECpH simulations to accompany sidechain reorientations of residues D54, H70, D73 and R147 that occur as a result of the changes in protonation states, as shown in Figure 1B,C. Moreover, the juxtaposition of HR-AFM maps of CLC-ec1 at pH 4.6 and 7.5 also revealed conformational rearrangements in the I–J loop, as evidenced by a decreased gap between AFM peaks attributed to residue K243 [19]. This ability of the ECpH approach to highlight the pH-dependent behavior of the all-atom CLC-ec1 system had eluded our conventional MD simulations in Ref. [19], pointing to the significant advantages of using partial protonation probabilities, over the binary representation of protonation changes in cMD. The results obtained for the CLC-ec1 transporter system demonstrate the capabilities of ECpH to distinguish in this manner between the effects of pH in the range of physiologically relevant values (5–8) with the updated set of individual pKa values of titratable residues estimated by PROPKAtraj [25,26], a substantial advantage over the cMD framework, which captures effects of a protein’s dynamics only in a pH range in which only histidine is expected to change its protonation state. 

## 4. Materials and Methods

### 4.1. Definition of the ECpH Framework

A comprehensive description of the ECpH methodology is presented in [20]. Here, we define the main parameters used in the MD of the CLC-ec1 molecular system with this method.

#### 4.1.1. Individual pKa Definition

To model the conformational transition to an open-gate conformation of the channel, molecular dynamics simulations were carried out by starting at both pH 4 and pH 8. The pKa values were estimated by applying PROPKATraj to the 40 × 500 ns trajectories of CLC-ec1 at pH 7 described in our previous publication [19].

#### 4.1.2. Protonation State Representation

The representation of ensemble protonation states is realized in the semi-grand canonical ensemble framework by the concept of co-existing protonation states. This is described by discrete switching between the protonation states of the titratable residues. Because of the high frequency of proton exchange, this approach requires significant computational resources, and for the real-world all-atom protein systems remains mostly unfeasible. For this reason, the ECpH method employs a force-field modification scheme. Our approach is formulated in the same *n*, *P*, *T*-ensemble framework that is commonly used in MD simulations (see Appendix B), and using the same information for input pKa values. The specific protonation scheme (Section 4.1.3) that enables the representation of the protonation states is a central element of the theoretical framework of the ECpH method. 

#### 4.1.3. The Protonation Scheme: Definition of The Effective Forcefield

To enable the shift between protonated and not protonated states of a molecular system composed of any number of protonatable sites, the number of particles is kept constant (no protons are added or subtracted), but the presence/absence of a proton at each site is then represented by linearly scaling the non-bonded interactions. This scaling by an individual protonation probability value is applied to sites with titratable hydrogens. The protonation probability is dynamically determined by (1) the ambient pH, and (2) the pKa value of that site in a particular conformational state of the entire system. To achieve a gradual shift between the protonated/non-protonated states of the system, the charges of the atoms neighboring the titratable residues are also scaled linearly, by the same scaling factor. Once individual pKa values are set for each protonatable residue in the system, the protonation probabilities are evaluated from the equilibrium constant of the protonation/deprotonation process with the Hill equation.

As presented in detail elsewhere [20], the core idea of the ECpH method is the introduction of the modified force field, representing the pH-dependence of the state of the system. A single canonical ensemble is created for each pH in a given range using this protonation scheme as the corresponding protonation probabilities modify accordingly the internal energies of the titratable protonation sites of the system (see Appendix B). We follow the common protocol [29,30] of applying scaling only to nonbonded terms of the potential energy to eliminate the dependence of kinetic energy on pH. 

The electrostatic potential energy and van der Waals terms for interactions of titratable protons are scaled by λ in Equations (1) and (2):(1)$$U_{elec}\left(\lambda ,r_{ij}\right)=\lambda \cdot \frac{q_{i}q_{j}}{4\pi \varepsilon_{0}r_{ij}}$$
(2)$$U_{steric}\left(\lambda ,r_{ij}\right)=\lambda \cdot 4\sigma_{ij}\cdot \left[{\left(\frac{1}{r_{ij}}\right)}^{12}-{\left(\frac{1}{r_{ij}}\right)}^{6}\right],$$
where $q_{i}$ and $q_{j}$ are the partial charges of the two atoms, $r_{ij}$ is the distance between them, and $\varepsilon_{0}$, $\sigma_{ij}$ are, respectively, the electrostatic constant and the Lennard-Jones well depth.

The partial charges of atoms neighboring the titratable residues, which are affected by the change in protonation state, are calculated as follows:(3)$$q_{current}\left(\lambda \right)=q_{prot}-\left(1-\lambda \right)\cdot \Delta q_{prot\to nonprot},$$
where $q_{current}$ represents the new value of the partial charges that correspond to the current level of pH, $q_{prot}$ stands for the partial charge value in the force-field for the atom in a protonated state, and $\Delta q_{prot\to nonprot}$ is the difference between the force-filed partial charges of the atom in a protonated and a non-protonated state. 

Amino acid residues considered as titratable sites include aspartate, glutamate, histidine, cysteine, and lysine [31]. Because histidine can be involved in two pH-dependent transitions—(i) deprotonation of the protonated state and (ii) transition between the ε and *δ* tautomeric forms—we use two scaling parameters in Equations (1)–(3) for the histidine: the *λ* to model deprotonation, and a binary parameter to encode the information about the tautomeric form. 

We note that because adaptation of partial charges of the titratable residues to the environmental pH leads to small changes in the system’s net charges, we have developed a neutralization protocol. This protocol distributes the excessive charge to the water molecules, mimicking the wet-lab titration experiment (see Appendix B).

### 4.2. Parameters Used in the cMD and ECpH Simulations 

All MD simulations (both ECpH and cMD) were performed with the OpenMM 7.5 software [32] using the CHARMM36 all-atom force field [33], in the NPT thermodynamic ensemble at P = 1 bar and T = 310 K. The pressure was preserved by either a Monte Carlo barostat or its membrane modification implemented in OpenMM (MonteCarloMembraneBarostat) [32]. Long-range electrostatic interactions were evaluated with PME. Nonbonded interactions’ cutoff and switching distances were set to 12 Å and 10 Å, respectively. The integration step was set to 2 fs, while the fluctuations of water bonds were constrained. The pKa values used in assigning the protonation states for titratable residues in both cMD and ECpH simulations were Glu: 4.4, Asp: 4.0, Lys: 10.4, His 6.5, 9.1, Cys: 9.5, unless stated differently (see below). Cys residues involved in disulfide bridges were not titrated.

#### 4.2.1. Individual pKa Assignment in CLC-ec1 Homodimer

For pKa estimation of the CLC-ec1 system, we have applied a recently developed extension to the PROPKA algorithm—PROPKA Traj [25,26]—to cMD trajectories of the protein obtained previously and described in detail in Section 4.2.2 and in ref. [19]. According to the estimated accuracy of PROPKA (RMSD = 0.89) [26], we have altered the model value of individual pKa for residues for which the median of PROPKA Traj-predicted distribution deviated from the model value by more than 1 pKa unit. If, on the other hand, the standard deviation of pKa prediction exceeded 1 pKa unit, the value of the pKa standard deviation itself was taken as a threshold instead. 

#### 4.2.2. ECpH and cMD Simulation Protocols for MD Trajectories of CLC-ec1 System

For CLC-eq1, the simulated molecular construct was built from PDB ID 1OTS [16] and was embedded in the membrane with the CHARMM-GUI Membrane Builder server [33,34,35]. The membrane was composed of 629 lipid molecules with a 70:30 composition of POPE:POPG. The system was solvated in explicit water solution with 0.15 M KCl. The equilibration procedure followed the standard CHARMM-GUI equilibration protocol in NAMD 2.10 software [36,37]. The ECpH MD simulation was performed for a pH range from 2 to 9 with step 1 in replicas of 6 for ~1 µs each. In both ECpH and cMD simulations published previously [19], pKa values for Glu113 and Glu148 were set at 8 and 6.5, respectively, according to experimental and computational data [38,39]. All other titratable residues were assigned the pKa values of the corresponding amino acids. For cMD of the CLC-ec1 system, data were generated from two parallel trajectories of 1.6 µs each for both pH 4.5 and pH 7.5 [19].

#### 4.2.3. ECpH and cMD Simulation Protocols for MD Trajectories of BBL Protein

For the ECpH simulations of the test BBL system described in Appendix C, the pH range was from 2 to 11, running for 1 µs at each pH value. The runs were initiated from both opened and closed states of the EF loop. The crystal structures of BBL at pH~8.1 (PDB ID 2BLG), and at ~6.3 (PDB ID 3BLG), were used as starting points. In all the simulations, the pKa value of Glu89 was set to 7.3 according to experimental data [40,41]. The corresponding cMD simulations were performed for pH 2, 6 and 8. At pH 2, all titratable residues were protonated, while ensembles at pH 6 and 8 differed in the protonation states of Glu89 and His residues. We also performed simulations starting from each of the two states of the EF loop, for 1 µs each. 

#### 4.2.4. Native Contacts Analysis Procedure

The analysis of native contacts was performed on 1 µs trajectories at a pH range from 2 to 11 for ECpH and pH 2, 6 and 8 for cMD. The distance cutoff was set to 5 Å.

## Figures and Tables

**Figure 1 molecules-26-06956-f001:**
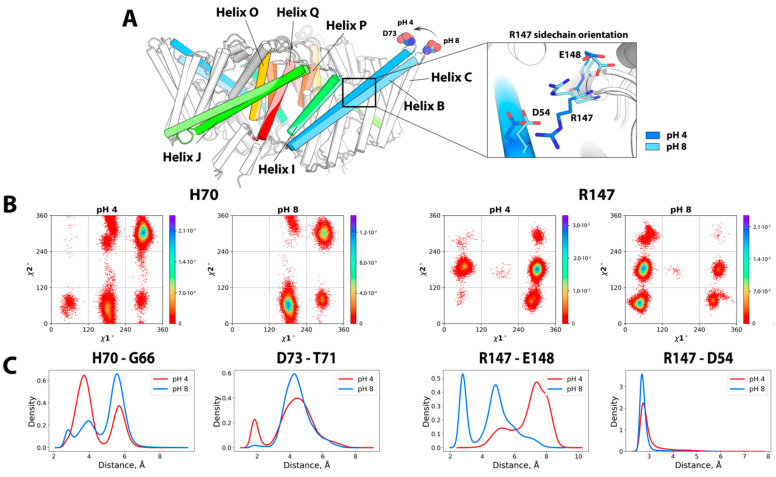
(**A**) The colored structural elements in the superposition of CLC-ec1 show the pH 4 form in the dark hues, and at pH 8 in the lighter hue of the same color. The zoom-in insert shows orientations of R147 at pH 4 (in dark blue) and at pH 8 (in light blue). (**B**) 2D histograms of the distribution of the dihedral angles χ^1^ and χ^2^ at pH 4 and 8 for H70 (left two panels) and R147 (right panels). (**C**) Kernel plots of density (probability) distribution for the H-bonds stabilizing the conformation between the sidechain of H70, D73 and the backbone oxygen of G66 and T71, (left two panels) and the sidechains of R147 and E148, D54 (right panels).

**Figure 2 molecules-26-06956-f002:**
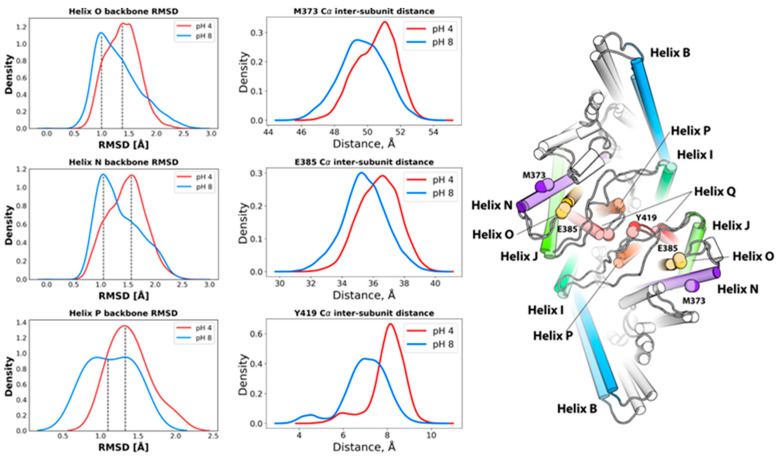
Displacement of helices N, O and P in ECpH simulations at pH 4. The first column of the plots shows the distributions of backbone atoms RMSDs for the helices relative to the starting conformation of CLC-ec1 at neutral pH. The second column presents the distributions of inter-subunit distances between the Ca atoms of M373 (N), E385 (O) and Y419 (P) in the ECpH trajectories at pH 4 (darker colors) and 8 (lighter colors). The structural representation on the right highlights the displacements of helices N, O and P.

**Figure 3 molecules-26-06956-f003:**
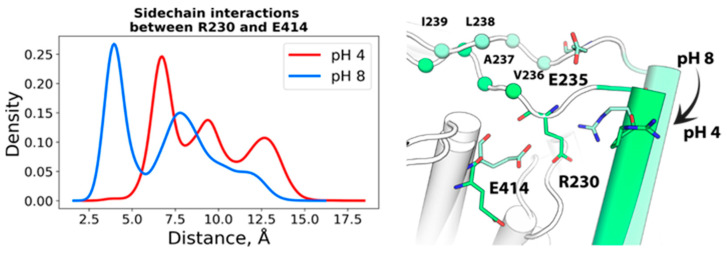
Left: distribution of minimum distance between sidechains of R230 and E414 at pH 4 and 8. Right: bending of I–J extracellular loop at pH 4.

## Data Availability

The ECpH python code compatible with OpenMM is published in GitHub: https://github.com/weinsteinlab/ECpH-MD. Along with the ECpH code the GitHub repository contains tutorial files for BBL protein system and step-by step instructions to perform the calculations reported in this manuscript.

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
