# Peer review of "Simulation of pH-Dependent Conformational Transitions in Membrane Proteins: The CLC-ec1 Cl/H+ Antiporter"

_molecules, 2021, doi:10.3390/molecules26226956_

Round 1

Reviewer 1 Report

Comments to manuscript ID - molecules-1458565:

  1. The statement “Remarkably, no major conformational rearrangements had been observed in experimental investigations of CLC-ec1 structure upon transition to a gate-opened state at pH 4.5 from physiological pH conditions.“ in the introduction would require a citation.
  2. ECpH method is in detail described in [20]. It would be, therefore, useful to limit its description in the main manuscript to particular parameters used in ECpH simulations. The rest can be moved to the supplementary material. The same applies to the BBL test system mentioned in appendix B.
  3. HSQC in appendix B is not “heteronuclear sequential quantum correlation” but “heteronuclear single quantum coherence”
  4. The title in paragraph 2.1.1 is “Spatial reorganization of helices A and B” however the discussion is about repositioning helices B and C. The helix C is not shown in Fig. 1A. The inset in Fig. 1A shows R147 sidechain reorientation but the colors are very similar and therefore the reorientation is not clearly visible.

Reviewer 2 Report

This is an interesting manuscript, clearly written and presented. Several important pieces of information are, however missing:

A) lines 134-135 state "Starting from the equilibrated structure, parallel ECpH simulations at pH 4 and 8 were run in equivalent swarms of replicas." How many replicas? How long was each replica? This must me clearly stated in the methods section.  Incidentally, the methods section (lines 318-319) states that pH from 2 to 9 were used instead. Please detail exactly how many replicates at each pH were perfomed.

B) In ECpH MD simulations, pKa E113=8 and pKa E148=6.5. This is crucial because it  makes a stable E148-R147 salt bridge  possible only above pH=6.5. Were the protonation states of these residues in the cMD chosen using these pKas, to ensure better comparability between cMD and ECpH MD?

C) in lines 225, authors state "These changes are seen from the ECpH simulations to result from switching the protonation states of D54, H70 and D73 residues". Such data are, however, not shown. This lack should be corrected. Please include graphs showing how (for example) salt-bridges or H-bonds involving these residues change as pH is varied. 

D) in section 4.2.1, it is stated "According to the estimated accuracy of PROPKA  (RMSD = 8.9) [25], we have altered the model value of individual pKa for residues that  exhibited pKa shifts >1 pKa unit with standard deviation of the value < 1 pKa unit." I do not understand what authors mean here. Please rephrase .Incidentally, I do not think RMSD has such a large RMSD: that would make it unusable and much worse than a method that simply assigned a pKa of 4.5 to every ASp/Glu and 6.8 to every His.

E) The graphs in Appendix B are not clear, and are at points insufficient for the claims  presented.  Specifically:

E.1) What do the straight lines in panels C Figure A1 mean,?

E.2) Figures A1 and A2 show evolution of RMSD, and the text uses those data to argue for "common trends of pH dependence". This is not correct, since two very different conformation may have the same RMSD relative to a common reference. Authors should instead show evolution of key multidimensional metrics (e.g. distance between aminoacids that are known to be close in the folded state, etc.)

E.3) In figure A.2, a broken line at RMS 4 angstrom is present and the legend states "X-ray at pH=6.2 (or 8.2)"  What is the reference used to compute that RMSD??

E.4) "We observed the opening of the EF-loop at pH 8 in 1 μs trajectories obtained from both methods (Figure A2, right)."  Fig. A2 is an RMSD graph, which cannot tell us whether (unlike the authors  stated) the EF-loop is opening or whether some other portion of the protein is unfolding.

E.5) Figure A3 has too many superposed points: one cannot ell, for example, whether in panel B there are any unfolded points at pH=3,4,5,6,8 , or whether any "intermediate" points at pH 2 are obscured by the light blue blob on the upper left quadrant.
